# Reporting and Interpreting Effect Sizes in Applied Health-Related Settings: The Case of Spirituality and Substance Abuse

**DOI:** 10.3390/healthcare11010133

**Published:** 2022-12-31

**Authors:** Iván Sánchez-Iglesias, Jesús Saiz, Antonio J. Molina, Tamara L. Goldsby

**Affiliations:** 1Department of Psychobiology & Behavioral Sciences Methods, Complutense University of Madrid, 28223 Madrid, Spain; 2Department of Social, Work and Differential Psychology, Complutense University of Madrid, 28223 Madrid, Spain; 3Department of Family Medicine and Public Health, University of California San Diego, San Diego, CA 92093, USA

**Keywords:** effect size, scientific writing, substance abuse, spiritually-based treatment

## Abstract

Inferential analysis using null hypothesis significance testing (NHST) allows accepting or rejecting a null hypothesis. Nevertheless, rejecting a null hypothesis and concluding there is a statistical effect does not provide a clue as to its practical relevance or magnitude. This process is key to assessing the effect size (ES) of significant results, be it using context (comparing the magnitude of the effect to similar studies or day-to-day effects) or statistical estimators, which also should be sufficiently interpreted. This is especially true in clinical settings, where decision-making affects patients’ lives. We carried out a systematic review for the years 2015 to 2020 utilizing Scopus, PubMed, and various ProQuest databases, searching for empirical research articles with inferential results linking spirituality to substance abuse outcomes. Out of the 19 studies selected, 11 (57.9%) reported no ES index, and 9 (47.4%) reported no interpretation of the magnitude or relevance of their findings. The results of this review, although limited to the area of substance abuse and spiritual interventions, are a cautionary tale for other research topics. Gauging and interpreting effect sizes contributes to a better understanding of the subject under scrutiny in any discipline.

## 1. Introduction

In any quantitative, scientific study, it is possible to accept or reject a null hypothesis using inferential analysis and null hypothesis significance testing (NHST). However, rejecting a null hypothesis and coming to the conclusion that there is a statistical effect does not indicate its practical relevance or the size of the effect. It is essential to determine the effect size (ES) of results that are statistically significant, either by using context (comparing the size of the effect to that of other studies or everyday effects) or by calculating statistical indices, which, in turn, must also be properly interpreted. This issue is relevant, especially in applied settings. There are multiple published recommendations encouraging authors to report (and interpret) ES indices. However, in many empirical studies using inferential statistics, such indices do not appear or are not adequately (or at all) interpreted. Not knowing the magnitude of the relationship between theoretically relevant variables or the effect of certain treatments may lead resource managers to make the wrong decisions. These decisions affect the health (physical or psychological) of individuals. We may risk spending too many resources on implementing a treatment that, although effective in practice, is not worthwhile when there are better or lower-cost alternatives. On the other hand, we might discard an intervention program as too costly, when its high effectiveness would make it worthwhile.

The aim of this article is twofold: on the one hand, to recall and insist on the importance of calculating and interpreting effect size (ES) indicators as part of inferential statistical analysis. On the other hand, to study the use of ES indicators in applied studies. Specifically, to illustrate this issue, we have chosen a health-related domain, that of substance abuse and its relationship with spirituality and religion.

We want to study to what extent authors of scientific articles calculate ES indices and whether (and how) they interpret them. Even in such a specific field, it will allow us to assess whether our perception of the problem is correct and whether it is necessary to insist on this issue. For this purpose, through a systematic review, we have selected empirical articles that relate substance abuse to spirituality (including those that assess intervention programs with a spiritual component). The discussion presented in this paper can be generalized, to some extent, to applied research in many fields. However, this research allows us to orient our recommendations on the use of ES indices to the context of substance abuse research, prevention, and treatment. This might be useful for applied researchers or therapists working in this field.

### 1.1. Scientific Research, Data Analysis, and Inferential Statistics

Scientific research in psychology relates to many other disciplines, such as epidemiology, biology, and medicine, among others. The research endeavor seeks to gain knowledge of human behavior in all of its aspects, from observable behavior to cognition, through personality traits, beliefs, attitudes, and many other systems and processes related to psychological or physical health. When psychological research addresses issues as prevalent as substance abuse, it becomes a public health issue. As with many other scientific disciplines, psychological research also seeks to describe, predict, and explain phenomena. The instruments needed to do so include a proper and thorough design and adequate data analysis to answer the proposed research question. Although there are certain alternative approaches to data analysis, such as Bayesian analyses, the most frequent strategy for inference is null hypothesis significance testing (NHST).

NHST is the key procedure in frequentist inferential statistics, while its use remains a subject of debate and controversy. Many of the criticisms [1,2,3] may be said to be based on misuse by researchers authoring studies and/or poor understanding on the part of both authors and readers [4,5,6]. The use of p-values is ubiquitous. Based upon the p-values, categorical, dichotomous judgments may be made regarding the so-called null hypothesis in terms of accepting or rejecting it. This in turn gives rise to a “significant” vs. “nonsignificant” results determination. Too often, that is the end of the road in a given study, and the authors draw conclusions on a substantive and complex issue from that *p*-value only. Usually, once an effect has been found, no attention is paid to the magnitude of that effect. Authors are just beginning to recommend the calculation and interpretation of the magnitude of an effect (ES) as part of what they refer to as “the new-statistics movement” [1,7,8]. However, as we shall see, the use of ES has been studied, discussed, and recommended as standard practice for decades now; at least, as far as we know, since 1969 [9].

### 1.2. Beyond the Null Hypothesis Significance Testing

Reporting ES does not replace the purpose of NHST (i.e., whether an observed effect is statistically significant or not), but supplies additional information regarding the magnitude of a significant observed effect (i.e., “How large an effect do I expect exists in the population?” [10]). The practical relevance of a given significant effect is better assessed by comparing it with reasonable criteria (well-stated effects in similar research settings, or everyday and well-known effects). Authors may call upon previous research in similar (if not the same) settings in order to compare ESs. If there are no contextual benchmarks, arbitrary but published thresholds are available for reference, even when they are not the best option. Labels such as “small”, “medium”, or “large”, may be misleading or simply uninformative as an ES estimator. It would seem that researchers use such labels, more often than not, to interpret ES indices as ubiquitous as Cohen’s *d*. However, Cohen himself, in 1988, believed that the convention he was proposing would be found to be “reasonable by reasonable people” [10] (p. 13) and warned about the dangers of the use and abuse of arbitrary labels.

Contrasting ES across different populations also assists in gaining knowledge of generalization strategies and identifying potential confounds affecting internal validity in any study. As Shadish et al. stated in 2002, “Demonstrating effect size variation across operations presumed to represent the same cause or effect can enhance external validity by showing that more constructs and causal relationships are involved than was originally envisaged; and in that case, it can eventually increase construct validity by preventing any mislabeling of the cause or effect inherent in the original choice of measures…” [11] (pp. 470–471).

ES may be interpreted using descriptive statistics only (that is, after a result has been deemed statistically significant and the sample statistics are to be interpreted). When variables are measured in units with intrinsic value (such as height or weight in standardized units) or contextual meaning (such as salaries in dollars), readers may make rapid assessments based on their previous experiences. The American Psychological Association (APA) [12] recommends including measures of effect size in the manuscripts and has been doing so since, at least, 2010. The APA mentions that ES expressed in original units allows for an easier interpretation (“e.g., mean number of questions answered correctly, kilograms per month for a regression slope”) [13] (p. 89), but focuses primarily on statistical estimates.

There are entire courses devoted to statistics in social sciences degrees. Descriptive and inferential statistics, psychometry, research methods, and epidemiology are subjects known to be taught in most (if not all) of those degrees, and ES indices are included in their syllabi. Additionally, there are numerous published papers that address this topic and offer recommendations regarding ES in several psychological research areas [14,15,16,17,18]. It is not clear whether the recommendations have been fully adopted over time by students, researchers, and reviewers alike. In fact, the misreporting (or lack of reporting) of ES remains an issue in scientific writing regarding several scientific disciplines and health-related settings. In a systematic review of randomized controlled trials (RCT) published since the year 2000, Martín-Aguilar and Sánchez-Iglesias [19] found that 8 out of 10 statistically significant studies (80%) failed to report ES statistics to assess the magnitude of the effect of pharmacological treatments on depressive symptoms; 1 reported ES statistics but did not interpret it; and only 1 reported and interpreted the ES in its context. In a similar review, Elvira-Flores and Sánchez-Iglesias [20] analyzed 21 experimental studies, 11 (52.4%) of which did not report ES statistics; 5 (23.8%) reported statistics but did not interpret them; and the remaining 5 (23.8%) reported and interpreted the ES values using the arbitrary thresholds proposed by Cohen [10] but without providing a contextual meaning.

### 1.3. Inferential Statistics without Effect Size Estimators and Questionable Research Practices

Failing to report ES indices, or doing so without discussing them, may be regarded as questionable research practices. Some reasons may include lack of training in statistical procedures, the rush for publishing imposed by competitive academic environments [21,22], a misunderstanding of the meaning and usefulness of ES, or a willingness to conceal observed poor effects. These practices may be found during statistical analyses, as in the case of ES calculations (or the lack of them) or other inadequate procedures (variable slicing, cherry picking, p-hacking, etc.). However, questionable practices may also occur before or after research [23], such as failing to publish non-significant results [24] or using tendentious causal language [25]. These questionable research practices pose a threat to the credibility of scientific research [26,27]. Not calculating effect sizes has the potential consequence of giving a more partial view of the phenomenon under study. Some statistical results (such as significance tests) then become a formal mathematical exercise that does not provide as much insight as it could. This makes the research a less useful and less-consulted resource.

We assume that, in most cases, the lack of ES reporting is unintentional. One may wonder whether these studies (and their results) may be considered “wrong”. We do not think so. However, even if the study was appropriately designed and reliable research methods were utilized, an argument may be made that they are not entirely complete. Readers will not have enough information to make more than educated guesses regarding the substantive relevance of the findings. Assuredly, readers may make their own assumptions regarding effect sizes and their interpretation. Assuming the relevant data (descriptive and inferential statistics) are reported, they may calculate ES indices and then interpret them. However, should it not be the authors who are the first to introduce and discuss the practical relevance of their own findings?

### 1.4. Spirituality, Religion, and Substance Abuse Studies

Psychological and social factors play a part in health problems. Religiousness is considered a key variable in health improvement [28,29,30], and researchers have studied spirituality and religiosity as relevant variables with regard to public health [31,32,33].

Religion is “an organized system of beliefs, practices, rituals, and symbols designed (a) to facilitate closeness to the sacred or transcendent (God, a higher power, or an ultimate truth/reality) and (b) to foster an understanding of one’s relationship and responsibility to others in living together in a community” [34] (p. 2). The term “spirituality” refers to a broader concept that encompasses everything from deeply religious people [32], to a characteristic of individuals who are only superficially religious, religion seekers, a well-being concept, or even secular individuals [35].

We focus the present study on substance abuse disorders. Recovery from other health and behavioral issues, such as gambling disorders (among other disorders), has been studied with regard to spiritual beliefs and practices [36,37]. Although they are occasionally classified as addictions, these issues are not directly related to substance usage and will not be addressed in the present study. The DSM-5 [38] recognizes substance use disorders as a pattern of troublesome symptoms resulting from substance use, from common substances such as alcohol or tobacco, to opioids, stimulants in general, and even other unidentified substances.

Addiction intervention programs are fundamentally divided into harm reduction and recovery-based programs. Harm reduction programs aim to minimize the main negative consequences of drug addiction [39], while recovery is a concept used to contextualize a process of treatment and subsequent social reintegration [40]. Recovery is occasionally used interchangeably with rehabilitation. However, the goal of rehabilitation is to assist individuals with a handicap or difficulty (such as an addiction) to reintegrate the individual into the community [41]. Recovery involves the development of personal autonomy and skills that allow socio-community integration and a relatively satisfactory life [42], and not only reducing or eliminating drug use, as in spontaneous remission [43,44].

Treatment networks currently include harm reduction programs, psychosocial integration programs, and recovery (or therapeutic) community programs [45]. Today, recovery communities feature empowerment, peer support, active participation, and social support [46]. Spirituality is an aspect that has received increased attention with regard to its role in the maintenance of recovery from alcoholism. Spirituality has been defined as that which gives meaning and purpose in life [47] as well as a sense of personal identity and transcendence that motivates individuals beyond the practicalities of daily living [48].

Recovery interventions, such as the Twelve-Step programs of Alcoholics Anonymous, advocate acceptance of a “higher power,” promote spiritual awakening, and use prayer and meditation as instruments for recovery and healing [49]. In this context, spirituality has been linked to betterment in certain health outcomes, including state anxiety in alcohol recovery [50] and relapse avoidance [51]. All these programs and interventions require a lot of resources, time dedication, and commitment on the part of the users. Knowing not only whether they are effective, but also to what extent they are effective, seems highly desirable. This would make it possible to choose which intervention is worthwhile for each individual, depending on his or her possibilities. On the other hand, as far as basic research is concerned, it is also useful to know to what extent characteristics related to spirituality are linked to substance abuse, its prevention, recovery, or relapse. This would allow the development of better prevention and intervention programs.

### 1.5. Objective

Using several databases, the present authors carried out a systematic review to obtain a non-biased sample of studies with inferential outcomes that linked spirituality or religion to substance abuse. The selected studies were analyzed to determine the number of studies that utilized ES estimators and/or interpreted the magnitude of their findings.

## 2. Methods

The systematic review procedure utilized in the present study was the PRISMA (Preferred Reporting Items for Systematic Reviews and Meta-Analyses) guidelines by Page et al. in 2021 [52].

### 2.1. Eligibility Criteria

In order to be included in the systematic review, the studies needed to be published between 2015 and 2020, in Spanish or English, and in a peer-reviewed scientific journal. The studies could use any methodology (experimental or not).

The target population was people who had a problem of substance abuse (any substance). For observational studies, the substance-related problem could have appeared at any time prior to the measurement of the variables.

In addition, the studies could be observational (assessing the relationship between spirituality and outcomes related to substance abuse) or include treatments, programs, or interventions based on spirituality or religion.

The studies had to present at least one significant outcome measure assessing the relationship between a relevant variable and a change in the abuse behavior, relapse prevention, or a theoretically related variable.

We excluded studies without significant outcomes, solely qualitative methodologies, or case reports.

### 2.2. Information Sources

The present authors carried out a systematic literature search, searching for relevant studies. The following ProQuest databases were utilized: PsycINFO, the Applied Social Sciences Index & Abstracts [ASSIA], Sociological Abstracts, and the Sociology Database (the latter three are included in the Sociology Collection), PubMed, and Scopus, for the period from 2015 to 2020.

### 2.3. Search Strategy

The same search terms were entered in each selected database, in English and Spanish, using the following Boolean expression: “(addiction OR “substance abuse”) AND (spirituality OR spiritual) AND (relapse OR treatment),” adapting the syntax to the specific rules of each database engine. The search was restricted by title, abstract, and keywords. The present authors also restricted the search to peer-reviewed papers published in scientific journals, excluding theses and dissertations, chapters, books, and gray literature items. The publication date was also restricted in the database, allowing registers from 2015 to 2020, both inclusive. The specific sequences of terms used for the ProQuest databases can be found in Appendix A.

### 2.4. Selection Process

In order to identify and remove duplicate records, we entered the data from the previous stage into a single Excel spreadsheet. To determine whether a record was suitable for retrieval and reading, two reviewers independently evaluated each record’s title and abstract. The final judgment was made with the assistance of a third researcher when appropriate. Disagreements among the reviewers were settled by consensus.

### 2.5. Data Collection Process

The present authors attempted to retrieve all eligible records. Two reviewers independently read these reports to determine their final inclusion and data extraction.

### 2.6. Data Items/Assessment of Effect Size Estimators and Their Interpretation

Each reviewer, on their own, searched for and extracted the methodology, statistical analysis techniques, ES estimators, and ES interpretations for each selected study. The ES estimators (contextual or statistical) were sought in the results section of each study. The reviewers also looked for effect size interpretations of the significant findings in both the results and discussion sections of each report. The studies were classified according to their methodology, main data analysis techniques, ES indices reported (explicitly as ES estimators or not), and the interpretation of the magnitude of the significant effects observed (again, explicitly reported as such or not). Disagreements were settled by consensus and with the aid of a third researcher, as in the previous step.

## 3. Results

### 3.1. Study Selection

We identified a total of 477 studies, and 294 non-duplicate records were screened. We excluded 268 records (241 by title and 27 by abstract); 26 were sought for retrieval and evaluated for eligibility. Of those, seven articles were excluded for the following reasons: The outcomes of two studies were non-significant, so ES was not necessary [53,54]; the outcome of one study was not related to change in substance abuse or improvement in relapse prevention [55]; the sample was not comprised of participants with a problem of substance abuse [56]; in another study, the intervention was not spiritually-based [57]; one did not report inferential statistics [58]; and one study could not be retrieved for full text [59]. Finally, 19 studies were included in the review. The flowchart of the search and selection of studies is displayed in Figure 1.

### 3.2. Study Characteristics

Table 1 shows the key characteristics of the selected studies. In summary, we identified the following designs in the 19 studies selected: cross-sectional, 6 studies (31.6%); longitudinal, 5 studies (26.3%); pre-experimental (one-group pretest-posttest design), 3 studies (15.8%); and 1 three-static, non-equivalent group design (5.3%). The remaining 4 studies used experimental or quasi-experimental designs (21.1%). A total of 12 studies (63.2%) included some kind of spiritually-related intervention.

The following is a summary of the selected articles, along with some comments on their treatment of effect sizes.

Abdollahi and Talib [60], in a cross-sectional study, examined the associations of several variables using a moderation test and structural equations modeling. The authors argued that spirituality and hardiness played a protective role against suicidal ideation (an abuse-related outcome) in a population with substance abuse referred to addiction treatment centers. They used the percentage of variance accounted for as an ES index with no benchmark or contextual comparison, stating that “hardiness and spirituality explain 46.0% of the variance in suicidal ideation. These findings indicate that hardiness and spirituality are valuable predictors of suicidal ideation (p. 17)”. However, to what extent are spirituality and hardiness protective factors compared to other factors? If their explained variance were greater than that of others, it might be convenient to enhance these traits (or if, being the smaller explained variance, enhancing them would be simple and costless).

Andó et al. [50] used path analysis to study the mediation effect of spirituality between anxiety and depressive symptoms and alcohol recovery in a three-static, non-equivalent group (three distinct alcohol treatment settings) design. They concluded that there is a beneficial effect of spirituality on decreasing state anxiety when attending long-term 12-step-based interventions, such as those provided by Alcoholics Anonymous. This study did not quantify the ES of these long-term interventions (and spirituality) on anxiety. The question remains as to how different the treatments are in terms of depression and state anxiety. We also need to know the strength of the association of spirituality as a trait with the effectiveness of 12-step interventions. Could an individual with low spirituality still benefit from this type of intervention?

Beckstead et al. [61] used a one-group pretest-posttest design to assess the change in young patients in a substance use treatment center after the incorporation of Dialectical Behavior Therapy, a spiritually-related treatment. They used a paired T-test to assess change, and Cohen’s d and its arbitrary benchmarks [62] to estimate the ES, stating that “The effect size of treatment, using Cohen’s d was 1.315, a large effect by Cohen’s standards” (p. 86). They also used arbitrary benchmarks to assess the ES of the percentage of change (clinically significant and reliable change on the YOQ-SR, a questionnaire designed to assess perceived functioning and distress). The authors reported that “...the clinical significance of change was substantial within individuals over time, with 96% of the youth either recovering or improving at the time of discharge (according to Jacobson & Truax, 1991 criteria)” [61] (p. 86). Although they had arbitrary thresholds, the clinical criteria are explained in contextual terms of the individual’s functioning. This allows the reader to get a sense of what was achieved by applying the treatment to that sample.

Crutchfield and Güss [63] designed a cross-sectional study examining a link between successful long-term substance abuse recovery and goal-oriented, educational, or vocational achievements. Their data analyses included T-test, Pearson’s correlation, and hierarchical linear regression. There was neither an explicit ES estimator for correlation values nor a contextual interpretation for R^2^ in the regression models. However, they used η^2^ as an ES estimator, using expressions such as “The magnitude of the differences in the means […] was large (eta squared = 0.12)” (p. 10). Moreover, they reported descriptive statistics and used them to express ES as a ratio in a meaningful metric, stating, “…This equates to roughly 10 years clean for those who said yes versus 5 years to those who said no” (p. 10). This ratio allows any reader to estimate how big the difference is between the two groups under consideration because the calendar metric is common and easily interpretable. However, the strength of association of the rest among the variables analyzed remains uninterpreted. Therefore, we do not know toward which achievements we should orient users with substance abuse problems, looking for the best long-term recovery.

Dickerson et al. [64], in a cross-sectional study with adults seeking substance use treatment, examined the relationship between several measures: demographic, mental health, physical health, cognitive functioning, cultural identity and spiritual involvement, and substance use-related variables. The authors found that higher frequency in traditional, spiritual practice correlated with lower depression and with lower generalized anxiety disorder scores. They used correlation analyses and reported *p*-values without r-values. This provides no indication, even if purely descriptive, of the strength of the relationship between the variables. Would it be worthwhile to spend resources on promoting traditional spiritual practices, with all the resources that this entails, as a community intervention?

Kelly and Eddie [65] used a cross-sectional, representative sample of adults who had had a problem with alcohol or drugs (AOD). They examined differences in spiritual and religious identification across groups, and whether those differences related to alcohol and other drug abuse recovery. Through chi-square analyses and post hoc tests, they found that spirituality (but not religiousness) related to recovery, but with some notable differences by ethnicity and gender. No ES estimators were calculated or discussed. The authors stated, “implications for including spiritual/religious concepts and linkages in treatment and recovery support service settings for Black Americans suffering from AOD problems” (p. 9). Knowing how much the presence of these spiritual aspects improves recovery is fundamental at knowing to what extent this conclusion is adequate.

Kerlin [66] found, in a one-group pretest-posttest design, a statistically significant decrease in self-reported health symptoms and therapeutic improvement as a result of a spiritually integrated treatment program for substance use disorder. She conducted multiple paired-sample *t*-tests on a sample of 30 women. However, the author did not report ES estimators, so the magnitude of the change could not be quantified. In the abstract, the author suggests clinicians “… to incorporate spirituality into treatment protocols, and/or encourage clients to join support groups that enhance spirituality”. This could be an unnecessary expenditure of resources if the change is trivial in practice.

Lashley [67] used a time series design to assess the impact of staying in a faith-based, addiction recovery program for homeless residents. She used paired *t*-tests to assess change and ANOVAs to compare differences based on demographic variables. The author found improvements in self-esteem, depressive symptomatology, and physical activity levels at follow-up periods after admission. No ES estimator was calculated, although some descriptive differences between groups were highlighted when reported in units with a contextual meaning. For example, the author stated, “On average, men reporting other religious affiliations having 54 fewer days in the program than men affiliated with the Christian religion (*p* < 0.05).” and “On average, men who had not used recovery resources in the past stayed nearly 67 days longer than men who had utilized past recovery resources…”. This is another example where a common metric for authors and readers makes it easy to put the differences found in context.

Lee et al. [68] performed a longitudinal study with youths diagnosed with substance dependency (alcohol and other drugs) in residential treatment with 12-step programs. They argued that this treatment played a role in promoting change. However, change was not quantified; no ES estimators were reported, although the authors used many statistical tools: Fisher’s exact test, Kruskal–Wallis chi-squared test, proportional hazard regression, binomial logistical regression, and random effects regression. Without indices and interpretations of ES, we cannot compare the observed change with that of other types of treatment or with spontaneous remission. This would allow us to choose the best type of treatment for young people.

Mallik et al. [69] designed a quasi-experimental study in which they compared the effects of spiritually-based meditation with relaxation and with standard treatment on substance abstinence, psychological distress, and psychological dysfunction. They concluded that the spiritually-based approach might add further support to substance use disorder patients. They used several statistical tests: ANOVA, chi-square test, logistic regression, ANCOVA, and moderation analysis. For the logistic regression, they used the odds ratio as an ES estimator and interpreted it in terms of likelihood ratio, e.g., “… participants in the Meditation condition were 22 times more likely to maintain abstinence than participants in the Relaxation condition and 15 times more likely to maintain abstinence than participants in the TAU condition” (p. 61). This allows a direct comparison between treatments, and its conclusion about the usefulness of the spiritually-informed approach as a supplement to other treatments (p. 63) is reasonably justified.

Medlock et al. [70] examined adult patients requiring medical detoxification for severe substance use disorders in another cross-sectional study. The researchers used bivariate analyses via Pearson’s correlation and multivariate linear regression models. In the former, no explicit ES measures (such as *R*^2^) were reported. They concluded that positive religious coping was negatively associated with days of substance use and positively associated with the use of mutual help. Furthermore, they associated religious coping with “… very modestly, yet statistically significantly lower craving” (p. 747), providing a clue regarding that specific ES. In the linear regression models, the change in *R*^2^ when including new variables to the model was mentioned, but not interpreted. The authors state that “Use of positive religious coping may modify the course of SUD recovery by promoting engagement in mutual-help activities” (p. 747). Implementing interventions to that effect would be interesting if it were easy, regardless of the size of the effect achieved. However, if it were costly, only its practical and relevant effects would justify its use.

Montes and Tonigan [71] administered measures longitudinally to a sample from community-based Alcoholics Anonymous (AA) and outpatient treatment programs. They examined spiritual and religious (S/R) practices as a mediator of the relationship between AA attendance and reductions in drinking behavior: They found this mediation effect (via mediation and moderated-mediation models) and concluded that some S/R practices should be fostered in order to positively change the drinking behavior. No ES index was calculated, however, some of the findings were placed into context, stating that “… the magnitude of the prognostic effect of gains in S/R practices on later increases in alcohol abstinence observed in the current study fell within the range explicated in a report by Tonigan (2015, in Magill et al., 2015)” (p. 8).

Ranes et al. [72] designed a longitudinal study with participants recruited from a 12-step-based residential program. Researchers asked participants to complete multiple instruments at baseline, the end of treatment, and three follow-up measures. They utilized repeated measures ANCOVA to assess changes in level of spirituality over time, while controlling for the effects of several variables. They also used multiple linear regression to evaluate predictive models, using R^2^ as an ES estimator but without further interpretation. The authors reported data plots and provided their opinion regarding the magnitude of the observed increment. For instance, they stated, “Data plots also demonstrated that spirituality increased throughout the duration of the study for all participants, with a large increase between baseline and the end of treatment” and “Participants with low baseline religiousness […] experienced a fairly large increase in spirituality during the first month following treatment” (p. 11). The authors conclude that their results provide support for the 12-Step model of treatment. However, knowing how much support we are talking about would allow us to compare this type of treatment with others, with different durations, required commitment, and associated costs.

Ransome et al. [73] studied religious involvement and race differences in opioid use disorder risk and found that religious involvement may be important for prevention and treatment practices. They utilized bivariate logistic regression to estimate the lifetime risk of opioid use disorder and data plots for visual interpretation of certain results; no explicit ES estimators were calculated or interpreted. Thus, their recommendation to clinicians to incorporate religiosity and spirituality into the treatment of opioid use disorders was not adequately justified in all cases.

Shorey et al. [74] considered mindfulness-based interventions promising as an effective intervention for improving substance use disorder and associated depressive symptoms. Using correlation and hierarchical regression analyses in a cross-sectional study, the researchers found that dispositional mindfulness and spirituality were negatively associated with depressive symptoms. They reported *R*^2^ and the change in *R*^2^ without further interpretation. They conclude that “…dispositional mindfulness was a robust predictor of depressive symptom clusters” (p. 342), but this statement is premature in the absence of evidence on the ES. This may lead to practical implications, such as incorporating mindfulness-based techniques without a clinically relevant effect.

Temme and Kopak [75] recruited participants from an inpatient residential therapeutic community. In an experimental design, they randomized the sample into an intervention group and a treatment as usual group. Using path analysis, the authors tested the model of relationships between mindfulness, spirituality (as a mediator), and warning signs of relapse. They did not report or interpret any ES estimators. The authors acknowledged that more research would be needed on how spirituality affects the recovery process and its importance in practice in order to make recommendations. Incorporating ES would have helped them address the second question.

Tianingrum et al. [76] designed a one-group pretest-posttest study and concluded that a Narcotics Anonymous-style intervention and rehabilitation may improve relapse prevention among prisoners with substance abuse problems. The authors used ANOVA and correlation analyses; however, they did not use ES indices, nor did they interpret the magnitude of their findings. The prison environment can afford to implement initiatives such as narcotics anonymous meetings and impose them on inmates. However, other and more effective initiatives could also be implemented; knowing which to choose depends on knowing the ES of interventions.

Yaghubi et al. [77] randomly assigned a sample of patients into two groups to evaluate the efficacy of religious-spiritual group therapy on the spiritual well-being and quality of life in methadone-treated patients, versus a no-treatment group. The authors found a significant increase in spiritual well-being for the experimental group using ANOVA, but they did not quantify the magnitude of the effect. In the context of this research, the authors consider religious –spiritual education to be an inexpensive and accessible resource. If so, the potential consequences of not assessing ES would be less severe. Their recommendation to apply it to patients would make sense even at low ES.

Yeterian et al. [78] studied religiosity and spirituality as predictors of cannabis use and heavy drinking, recruiting a sample of adolescents in outpatient treatment. The researchers randomly assigned the sample to a Twelve-Step Facilitation treatment group or to a Motivational/Cognitive-Behavioral Therapy group. The data were analyzed via correlation, hierarchical multiple linear regression, and logistic regression. ES were reported for linear regression (change in R2), but not interpreted. For logistic regression, the authors interpreted ES using the odd ratio in terms of an increase or decrease in the likelihood of a behavior; for instance, “For each 1-point decrease on the STS [Spiritual Transcendence Scale] at baseline, individuals were 3.34 times more likely to report HDD [heavy drinking day] at follow-up (i.e., 1/OR = 1/0.299 = 3.34).” (p. 6). It would have been better if it had been put in comparison with the findings of other studies. However, this type of interpretation allows us to get a sense of what happens as spirituality increases.

Altogether, in the 19 studies selected, the authors reported results from 42 main statistical techniques. The analyses were quite varied. They included descriptive statistics and graphical plots, chi-square and Fisher’s exact tests, independent and paired t-tests, ANOVAs and Kruskal–Wallis tests, ANCOVA, Pearson’s correlations, regression models (linear, hierarchical, logistic, random effects, and hazard regression), path analysis, moderation tests, and structural equation models. Of these 42 techniques, 29 (69.0%) did not include any ES index. The 13 ES estimators reported were: *R*^2^ (and/or change in *R*^2^), six times; odds ratio, two times; ratio expressed in a contextual frame, one time; *r*-value, one time; eta squared (η^2^), one time; Cohen’s *d*, one time; and percentage of change in a test scoring, one time. Out of 19 studies, 11 (57.9%) did not report any ES index at all.

ES interpretations were found on 12 occasions. Three indices were interpreted using arbitrary benchmarks (for Cohen’s *d*, η^2^, and Jacobson and Truax criteria [62]). Two ES indices (both from a single study) were interpreted as mean differences in a natural context (days). Two ES indices (both odds ratios) were interpreted as likelihood ratios. Another ES was interpreted as a ratio of years between two distinct groups. In one study, the authors did not report ES indices, but they put the results into context by comparing them to a similar study by other researchers. One ES estimator, expressed as “percentage of variance accounted for” was interpreted arbitrarily, without benchmarks or contextual framing. In the last two studies in which the magnitude of an effect was addressed, the authors did not report ES indices and they used subjective judgments or opinions. Out of 19 studies, 9 (47.4%) did not report any interpretation of the magnitude or relevance of their findings at all.

## 4. Discussion

This paper addresses the importance of properly gauging the magnitude of effects inferred via Null hypothesis significance testing (NHST). The present authors chose an unbiased sample of articles, thanks to a systematic review, to illustrate the need for better reporting of ESs in the applied field of substance abuse disorder interventions. Using the dichotomous decision method of NHST, we can test whether empirical data conform to a null model (as suggested by Fisher in 1925 [79]) or to an alternative one (as proposed in 1928 by Neyman and Pearson [80]). Both proposals were combined into the ubiquitous NHST. However, NHST was never intended for inferring clinical significance from statistical significance. Since then, multiple effect size (ES) indices have been proposed and widely used. There are primers and guides for using ES indices published elsewhere [14,15,16,17,18]. In addition, authors are encouraged to interpret ES within a contextual framework. Despite recommendations for estimating and interpreting ES, as in other research fields [19,20], the authors of the present paper found that studies of spiritual treatments in substance abuse patients rarely report any statistical index or any other type of estimator. Interpretations of the estimators are also infrequent, and when they do occur, they are mostly arbitrary thresholds using the “small”, “medium”, and “large” labels. Contextual references are very rare. As reported by other authors [81], we also found instances of interpretations left to the subjective judgment of the author. In the present study, approximately half of the selected studies did not report any ES index, and roughly the same number did not interpret the magnitude or relevance of their findings either.

In the Results, we can see that, in general, spirituality and treatments that include spiritual components are related to different aspects of recovery from substance abuse. Nevertheless, the findings of this work also lead us to the opinion that it would be useful to revisit and validate the relevance of recovery-based programs [82]. It is important to develop theoretical models and useful interventions based on scientific evidence, with data gathered in applied studies with people who have problems with addictive behaviors [83]. However, the practical relevance of a given intervention cannot be assessed, even if it yielded significant differences with a control group or a treatment-as-usual group, without gauging the magnitude of the differences (i.e., estimating and interpreting the ES). It is through the effect size that we will know whether, in the applied context of the research, the intervention is worthwhile. Furthermore, the qualitative interpretation of ES indices should not be carried out using arbitrary labels, offered uncritically. More often than not, the arbitrary label “large” is associated with the great importance of a phenomenon, whereas “small” leads to lukewarm or dismissive language [84]. Small ESs may be relevant if they can be obtained with short, simple, inexpensive intervention programs, or a combination of the above. Large ESs may determine that even the most expensive and complex intervention programs will be implemented. These are decisions that need to be made by hospital, institutional, or government managers. It is up to us, the researchers, to calculate and provide clear and accurate indications of the ES. It is we, from academia and applied research, who have the duty to report adequately on this fundamental aspect. Statistical indicators should not be a straitjacket for interpreting effect sizes, using strict thresholds and benchmarks with arbitrary meanings. Even after calculating indices such as Cohen’s *d* or *r*^2^, researchers need to interpret them in the framework of the actual context of the research. For instance, the standardized differences obtained from intervention and control groups can be compared between similar studies A and B. Is one of the interventions relatively better than the other? This comparison would be even better if, instead of the *d* indices of each study, we compared the scores on an interpretable metric. For example, the mean difference (between the experimental and control groups) of days elapsed without relapse in study A compared to that in study B. As we have found in this review, ratios [63], odd ratios, and likelihood ratios [69,78] are also statistics susceptible to straightforward contextual interpretation.

All this is especially true in clinical settings, where decision-making affects patients’ lives. The choice of an effective intervention is of paramount importance in substance abuse programs. The literature presents promising data on the inclusion of spirituality in recovery-oriented programs when it comes to treatment, relapse prevention, and social integration (particularly those emphasizing social support and recovery capital, participatory activity, and a biopsychosocial perspective) [45,46]. However, much information is lacking regarding how great its effect is compared to that of other types of treatment (or, indeed, the same type of treatment in different populations and conditions). Any comprehensive network for treating addictive behaviors should contain programs based on previously verified data, and the ES is essential to gauge their usefulness.

### 4.1. Limitations

This study is not without limitations. Firstly, the present study assessed the methodological rigor when reporting and interpreting ES for a very specific setting, spiritual-based interventions and programs, and their effect on recovery from substance abuse. Therefore, the search terms used were limited; the search may have been conducted with a more comprehensive search equation. Secondly, the eligibility criteria excluded purely qualitative studies. Many qualitative studies propose a priori hypotheses, and about half subject them to statistical tests [85]. In this sense, our review leaves out qualitative studies that could address the estimation of effect sizes, and future reviews could include this methodology. In qualitative studies that do not include statistical tests, the effect sizes are treated and conceived differently. For instance, researchers can enhance the hermeneutic process in a thematic analysis by quantifying the frequency of emergent themes, thus weighting the relative importance of each theme [86]. This approach departs from the quantitative conception of effect sizes of the present paper; however, it would be an interesting work in itself. Thirdly, our search led the present authors to a relatively small number of studies (*n* = 19) that we considered suitable according to the current study’s proposed inclusion and exclusion criteria. To acquire a larger sample of publications, an option might have been to conduct the search using various criteria, such as more databases, a wider range of publication dates, synonymous search phrases, etc. Moreover, one option might have been to search for articles on the impact of spiritual therapies on various aspects of health to gain access to a broader sample. Nevertheless, the primary goal was to discuss the importance of having a measure of the magnitude of the effects found in spiritual treatments for substance abuse. A non-biased selection of articles was obtained through the systematic review process. This non-biased sample allows us to assess how well the ES is addressed by the publishing authors. In addition, we do not suspect that the manner in which our research topic was approached by researchers differs from that of other studies. However, given the limitations of the size of our sample of studies, caution should be used when generalizing. Future studies could utilize a new applied research question to address this objective.

### 4.2. Conclusions

In this paper, a systematic review was conducted on a very specific health-related issue to highlight this argument in an applied setting of interest. The present research revealed that approximately half of the studies did not report effect size indicators. In addition, approximately half of the studies do not interpret effect size in any way. There is a promising body of research demonstrating the usefulness of spiritual therapies in treating health conditions, including substance abuse relapse. However, there is a need for improved methodological rigor when reporting and interpreting effect sizes. It is not only desirable to calculate and report statistical indicators, but also to place them in the context of the research. It could be argued that research on spiritual or religious interventions in substance abuse is not representative of general scientific research. However, the authors writing on this specific topic do not necessarily report their findings differently from other researchers. Thus, the present authors argue that the results of the current review stand as a cautionary tale, a warning for researchers in any area of applied research.

## Figures and Tables

**Figure 1 healthcare-11-00133-f001:**
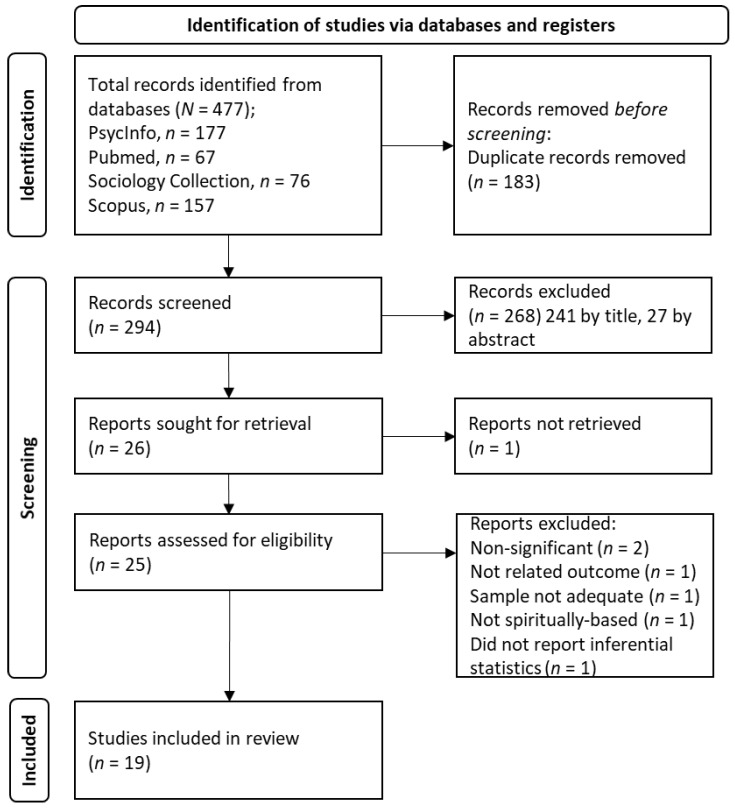
Flow Diagram of the Search and Selection of Studies.

**Table 1 healthcare-11-00133-t001:** Design, Main Statistical Analyses and Effect Size (ES) Interpretations in the Studies Selected.

Citation	Methodology	Statistical Analysis	ES	ES Interpretation
Abdollahi and Talib (2015)	Cross-sectional	Structural Model	Variance accounted for (%)	Arbitrary, no benchmark or context.
		Moderation Test via SEM	No	-
Andó et al. (2016)	Three static, non-equivalent groups design *	Path analysis	No	-
Beckstead et al. (2015)	Pre-experimental (one-group pretest-posttest design) *	T-test	Cohen’s *d*	Arbitrary benchmarks
		Descriptive statistics	% of clinically significant change	Arbitrary benchmarks
Crutchfield and Güss (2018)	Cross-sectional	T-test	η^2^	Arbitrary benchmarks
		Descriptive statistics	Ratio	Natural context
		Pearson’s correlation	*r*	-
		Hierarchical linear regression	*R* ^2^	-
Dickerson et al. (2021)	Cross-sectional	Correlation (w/o *r* value, only *p*-value)	No	-
Kelly and Eddie (2020)	Cross-sectional	Chi-square analyses, *post hoc* tests	No	-
Kerlin (2017)	Pre-experimental (one-group pretest-posttest design) *	Paired and independent T-tests	No	-
Lashley (2018)	Longitudinal *	Paired T-tests	No	Difference in mean (days)
		Correlation (w/o *r* value, only figures)	No	-
		ANOVA	No	Difference in mean (days)
Lee et al. (2017)	Longitudinal *	Fisher’s exact test	No	-
		Kruskal–Wallis chi-squared test	No	-
		Proportional hazard regression	No	-
		Binomial logistical regression	No	-
		Random effects regression	No	-
Mallik et al. (2019)	Quasi-experimental *	ANOVA	No	-
		Chi-square test	No	-
		Logistic regression	Odds ratio	Likelihood ratio
		ANCOVA	No	-
		Moderation analysis	No	-
Medlock et al. (2017)	Cross-sectional	Correlation	No	Subjective judgment
		Multivariable linear regression	Δ*R*^2^	-
Montes and Tonigan (2017)	Longitudinal *	Mediation and moderated-mediation	No	Context (similar studies)
Ranes et al. (2016)	Longitudinal *	ANCOVA	No	-
		Multiple linear regression	*R* ^2^	-
		Data plots	No	Subjective judgment
Ransome et al. (2019)	Longitudinal	Logistic regression	No	-
		Data plots	No	-
Shorey et al. (2015)	Cross-sectional	Correlation	No	-
		Hierarchical linear regression	*R*^2^ and Δ*R*^2^	-
Temme and Kopak (2016)	Experimental *	Path analysis	No	-
Tianingrum et al. (2018)	Pre-experimental (one-group pretest-posttest design) *	ANOVA	No	-
		Correlation	No	-
Yaghubi et al. (2019)	Experimental *	ANOVA	No	-
Yeterian et al. (2018)	Experimental *	Correlation	No	-
		Hierarchical linear regression	Δ*R*^2^	-
		Logistic regression	Odds ratio	Likelihood ratio

Note. SEM: Structural equation modeling. * This design comprised a spiritually-based intervention.

## Data Availability

Not applicable.

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
