# Peer review of "Reporting and Interpreting Effect Sizes in Applied Health-Related Settings: The Case of Spirituality and Substance Abuse"

_healthcare, 2022, doi:10.3390/healthcare11010133_

Round 1

Reviewer 1 Report

We read with great interest the article by Sánchez-Iglesias et al entitled “Reporting and Interpreting Effect Sizes in Applied Health- Related Settings: The Case of Spirituality and Substance Abuse” where the authors would apply “Inferential analysis using null hypothesis significance testing” (NHST) to test the contribution of effect size (ES) on data significance to be implemented into clinical studies which rely mainly on the p-value. The aim work has a major statistical implication in using calculation and interpretation of the magnitude of an effect (Effect Size, ES) as part of statistical analysis.

The work is well-written and has high experimental application there are a few minor  points that the authors need to comment on:

reporting ES indices has been applied in several studies, although many are descriptively labeled such as “small”, “medium”, or “large” rather than quantitatively computed, nevertheless, these studies are able to achieve the proper experimental endpoint. This also applies to both experimental and clinical studies as well. I would suggest that the authors would provide a roadmap to explain what necessitates the use of ES computation along with other statistical approaches and how it can be affecting results in different disciplines psychiatry, drug therapy, cancer, etc… ).

These points need to be added in the discussion section and the inclusion of the limitations section would be of importance where the authors can explain to the readers when certain studies are qualitative in nature and would not include or rely on ES and how one can overcome such studies.

Author Response

We would like to thank the editor and the two reviewers for the chance to submit an improved version of the manuscript. We have corrected some typos and grammatical errors, and have added missing information and clarified some issues.

Reviewer 1

We read with great interest the article by Sánchez-Iglesias et al entitled “Reporting and Interpreting Effect Sizes in Applied Health- Related Settings: The Case of Spirituality and Substance Abuse” where the authors would apply “Inferential analysis using null hypothesis significance testing” (NHST) to test the contribution of effect size (ES) on data significance to be implemented into clinical studies which rely mainly on the p-value. The aim work has a major statistical implication in using calculation and interpretation of the magnitude of an effect (Effect Size, ES) as part of statistical analysis.

The work is well-written and has high experimental application there are a few minor points that the authors need to comment on:

Thank you for your kind words and insights. In the new version of the manuscript, we have colored the added text green, in response to your recommendations and those of the other reviewer.

reporting ES indices has been applied in several studies, although many are descriptively labeled such as “small”, “medium”, or “large” rather than quantitatively computed, nevertheless, these studies are able to achieve the proper experimental endpoint. This also applies to both experimental and clinical studies as well.

The labels “small”, “medium”, or “large” are arbitrary thresholds that depend on vague descriptions that could be different between disciplines, or even applied settings within a specific discipline. Our criticism of these labels has to do with this vagueness. For instance, a "medium" effect size is only medium when it can be located among other studies where larger and smaller effect sizes have been found. If these studies are known, why not interpret the estimator by putting it in context? And, if they are not known, and therefore we cannot know if a certain estimator is "medium", why not report the numerical value? This value would also have no contextual interpretation but, at least, it would indicate more precisely what has been observed in the sample. We have added this line of reasoning in the section “Beyond the Null Hypothesis Significance Testing”.

I would suggest that the authors would provide a roadmap to explain what necessitates the use of ES computation along with other statistical approaches and how it can be affecting results in different disciplines psychiatry, drug therapy, cancer, etc… ).

We have tried to explain, in a generic way, how the use and interpretation of HE indices affects research, and we have illustrated this with spirituality and substance abuse. Thanks to the reviewers' comments we have put some more comments on this. Perhaps it would be too long (the article is already very long) to open the examples to other specific disciplines. On the other hand, creating a guide on the use of the different effect size indices would be another article in itself. The aim of this work was to check whether the recommendations in this respect are being complied with, and to recall the importance of doing so. We have cited other existing guides, and we do not rule out making our own guide in the future, but for this article it does not seem feasible.

These points need to be added in the discussion section and the inclusion of the limitations section would be of importance where the authors can explain to the readers when certain studies are qualitative in nature and would not include or rely on ES and how one can overcome such studies.

We have added information on effect size estimates in qualitative studies (those that use statistical tests and those that do not) in the limitations section.

Reviewer 2 Report

Thank you for the opportunity to revise an interesting piece of research. The authors read a lot, and prepared a good quality manuscript. I have, however, a number of comments and suggestions. I hope that by addressing them, the authors will improve their work and make it publishable. 

Introduction:

Recommendation 1. I would recommend reorganizing this section. Usually, the introduction should provide clear and concise answer to four questions, as follows: (1) what it is going to be studied, (2) why is the topic important, (3) what is the methodology to be employed and (4) what is the research gap to be filled, and what are the implications of the research. 

The authors provide answer to all these questions, but the aim of the study becomes available only on page 4. The rest of the answers are drown in too much information that does not belong to the introduction.

My suggestion is to start with a shorter introduction answering to the above mentioned questions within one page, and only then start building all the theoretical background you provide now. The introduction should grab the attention and provide concise information about the paper.

Recommendation 2. The "conflict" in your research goal is not properly emphasized. A lot of time is devoted to discussions about each concept involved (mainly definitions), but this first part should better pinpoint the research gap you intend to fill and the importance of not having it filled. 

As it is now, your paper starts with the idea that ES should be reported and interpreted (and I fully agree with this), and shows that among the 19 articles identified as related to spirituality and substance abuse, only a few did what they were supposed to do. In my opinion, this is good but it's just descriptive statistic. What are the potential consequences of not reporting the ES in the papers you included in your final analysis? What are the costs? Is there anything at stake? Please review the description of the papers (the Results section) having in mind the effects of not including references to ES. Otherwise, your systematic review is nothing but a list of resumes. 

Recommendation 3. Section 1.2. - in my opinion, this section should provide information about potential consequences of not reporting ES, and not a list of motives for not reporting them. Unless researchers understand the effects of not focusing on this dimension, they will keep ignoring it. 

Recommendation 4. Section 1.3. - the content does not match the title. The section is a long sequence of definitions, but nothing about the importance of the ES in this area. Please, reconsider. 

Recommendation 5. Section 1.4. - you mention "several debates" on line 192. Please explain and use these debates to better formulate your research aim and identify your research gap. 

Results:

Recommendation 6. You provide a nice summary of the main content of each paper identified as suitable for your research. However, like I mentioned above, there is nothing about potential consequences of not reporting ES, or reporting it incompletely. You must explain why this missing information may harm the research, and provide references accordingly. 

Discussions:

Recommendation 7. Avoid the word “believe” in a scientific research.

Recommendation 8. The discussions seem to be unrelated with the results described in the previous section. They provide general information, and are hardly built on the specific concepts of interest - the relation between spirituality and substance abuse. 

*

At the end of my report I want to congratulate the authors on an interesting research, and express my belief that after reorganizing the information, exclude some unnecessary definitions and discuss the 19 identified papers from a broader perspective, the manuscript can be suitable for publication. 

Author Response

We would like to thank the editor and the two reviewers for the chance to submit an improved version of the manuscript. We have corrected some typos and grammatical errors, and have added missing information and clarified some issues.

Reviewer 2

Thank you for the opportunity to revise an interesting piece of research. The authors read a lot, and prepared a good quality manuscript. I have, however, a number of comments and suggestions. I hope that by addressing them, the authors will improve their work and make it publishable.

Thank you for your keen eye and thoughtful comments. We will try to follow your recommendations. We really believe that this will improve the reading. In the new version of the manuscript, we have colored the added text green, in response to your recommendations and those of the other reviewer.

Introduction:

Recommendation 1. I would recommend reorganizing this section. Usually, the introduction should provide clear and concise answer to four questions, as follows: (1) what it is going to be studied, (2) why is the topic important, (3) what is the methodology to be employed and (4) what is the research gap to be filled, and what are the implications of the research.

The authors provide answer to all these questions, but the aim of the study becomes available only on page 4. The rest of the answers are drown in too much information that does not belong to the introduction.

My suggestion is to start with a shorter introduction answering to the above mentioned questions within one page, and only then start building all the theoretical background you provide now. The introduction should grab the attention and provide concise information about the paper.

Following this recommendation will allow us to more easily capture the reader's attention. We tried to answer the four questions at the beginning of the introduction. We have renumbered the sections of the introduction to accommodate this new information.

Recommendation 2. The "conflict" in your research goal is not properly emphasized. A lot of time is devoted to discussions about each concept involved (mainly definitions), but this first part should better pinpoint the research gap you intend to fill and the importance of not having it filled.

In the first part of the introduction, we have also tried to highlight this in a concise manner.

As it is now, your paper starts with the idea that ES should be reported and interpreted (and I fully agree with this), and shows that among the 19 articles identified as related to spirituality and substance abuse, only a few did what they were supposed to do. In my opinion, this is good but it's just descriptive statistic. What are the potential consequences of not reporting the ES in the papers you included in your final analysis? What are the costs? Is there anything at stake? Please review the description of the papers (the Results section) having in mind the effects of not including references to ES. Otherwise, your systematic review is nothing but a list of resumes.

We have kept the description of the studies as it is in Results, adding some comments on their treatment (or lack of it) of ES.

Recommendation 3. Section 1.2. - in my opinion, this section should provide information about potential consequences of not reporting ES, and not a list of motives for not reporting them. Unless researchers understand the effects of not focusing on this dimension, they will keep ignoring it.

We have added some comments on that in Section 1.2 (now numbered 1.3). In addition, we have added a brief commentary on a practical benefit (related to the publication process) of reporting and interpreting ES indices.

Recommendation 4. Section 1.3. - the content does not match the title. The section is a long sequence of definitions, but nothing about the importance of the ES in this area. Please, reconsider.

You are right; the content does not match the title. The title of this section was retained from a previous version where there was content related to ES in the field of spirituality, Religion, and substance Abuse. Now is entitled simply “Spirituality, Religion, and Substance Abuse Studies” (even though we now end the section by briefly discussing the usefulness of knowing the effect size of the relationship between these variables). We also agree that this section (now numbered 1.4) was too long, with too many definitions. We have shortened it.

Recommendation 5. Section 1.4. - you mention "several debates" on line 192. Please explain and use these debates to better formulate your research aim and identify your research gap.

Section 1.4. (now numbered 1.5) begins by mentioning that we used bibliographic databases in the systematic review. Perhaps you read debates instead of databases?

Results:

Recommendation 6. You provide a nice summary of the main content of each paper identified as suitable for your research. However, like I mentioned above, there is nothing about potential consequences of not reporting ES, or reporting it incompletely. You must explain why this missing information may harm the research, and provide references accordingly.

We have added comments on that along the Results section, following Recommendation 2.

Discussions:

Recommendation 7. Avoid the word “believe” in a scientific research.

We used the term “believe” as the following statement (“…it would be useful to revisit and validate the relevance of recovery-based programs”) was not a direct result of the findings of our review. However, it is somewhat related to them, so it is convenient to use another expression. We have changed it to “The findings of this work lead us to the opinion that it would be useful to…”

Recommendation 8. The discussions seem to be unrelated with the results described in the previous section. They provide general information, and are hardly built on the specific concepts of interest - the relation between spirituality and substance abuse.

In fact, the general objective of the article is more along the lines of providing general information on effect size that can be applied in different research settings. Substance abuse and its relationship with spirituality is an applied field to illustrate this methodological aspect, and to be able to evaluate its compliance in scientific publications. We believe that this is clearer now at the very beginning of the introduction. However, there is now more information on this applied aspect (mostly in Results, but also some more in Discussion), perhaps the discussion is more related to the findings.

*At the end of my report I want to congratulate the authors on an interesting research, and express my belief that after reorganizing the information, exclude some unnecessary definitions and discuss the 19 identified papers from a broader perspective, the manuscript can be suitable for publication.

Thank you again for your recommendations and your kind words. We hope that the changes we have implemented, along with those related to the other reviewer's comments, have improved the manuscript.

Round 2

Reviewer 2 Report

In general I am fine with how the authors responded to my comments and suggestions, therefore I will recommend acceptance provided two minor suggestions.

First, I would not start the paper stating the aim. It looks too... straightforward. I would provide a bit of context and only then I would explain the aim. But sure, this is my perspective. Most likely the authors organized the introduction the way they did because of my previous review report where I referred to the four questions to be answered: what, why, how, and gap to be filled. However, the answers are not supposed to be found in this order.

Second, I would give a second thought to what is written on lines  88-93 and 144-152. I am a social scientist and I know that in my field effect sizes are important mainly because they speak on the relevance to pursue practical interventions. And this should be valid in health research too. The relation between substance abuse and anxiety (for instance) can be statistically significant, but if the corresponding effect size is too small I know that by acting upon substance abuse I would not get the anxiety decreased.

Effect size is also useful in regression models to rank between predictors and see which one is best suitable for interventions. Although many regression relationships can be statistically significant, although some of them can have effect sizes suitable for intervention, there is always a rank among predictors that points towards the best to be considered. 

From this perspective, I question the authors' argument that not reporting effect size has effects on publication time. Although it may have such an effect, a statement like this may signal that the authors are not aware of the true consequences of not reporting their measure of interest.

(In fact, in my personal opinion the under reporting of the effect sizes in the extant literature comes from authors who don't understand the practical relevance of this measure.)

In addition, in my field there are theoretical thresholds available, depending on the type of effect size employed. For instance, in PLS-SEM we use a Cohen-like effect size with a 0.02 minimum threshold, and several intermediary values to delimitate between small, medium and large. I trust that in health research there are thresholds too, therefore I would try to take a more in depth look in the first paragraph of section 1.2 and maybe make it more concrete. 

Author Response

Response to reviewer 2

In general I am fine with how the authors responded to my comments and suggestions, therefore I will recommend acceptance provided two minor suggestions.

We would like to thank the reviewer once again for taking the time to provide thoughtful comments on the manuscript.

First, I would not start the paper stating the aim. It looks too... straightforward. I would provide a bit of context and only then I would explain the aim. But sure, this is my perspective. Most likely the authors organized the introduction the way they did because of my previous review report where I referred to the four questions to be answered: what, why, how, and gap to be filled. However, the answers are not supposed to be found in this order.

Indeed, we adjusted the beginning of the introduction taking into account your comment in the previous review. This time we have reorganized the information, with the same objective (to start the manuscript with a shorter introduction answering to the four questions, within one page), but not as directly. We begin with a very concise background information about NHST (adding a few lines, colored green), then stating the problem as we see it, then the aim of the paper, and then how we want to make it.

Second, I would give a second thought to what is written on lines  88-93 and 144-152. I am a social scientist and I know that in my field effect sizes are important mainly because they speak on the relevance to pursue practical interventions. And this should be valid in health research too. The relation between substance abuse and anxiety (for instance) can be statistically significant, but if the corresponding effect size is too small I know that by acting upon substance abuse I would not get the anxiety decreased.

Effect size is also useful in regression models to rank between predictors and see which one is best suitable for interventions. Although many regression relationships can be statistically significant, although some of them can have effect sizes suitable for intervention, there is always a rank among predictors that points towards the best to be considered.

Regarding lines 88-93, what we mean is that there must be something to compare to (a benchmark, if you will) in order to choose the adjective to be used. This is compatible with what you commented about regression models. Within the same model, we can compare the effect size of a predictor with those of the other predictors included. Then, using the context (the difficulty of the practical intervention on that independent variable to produce changes in the dependent variable) and the magnitudes of the effects found, we could choose the most appropriate predictors on which to intervene. In the interpretation of such a model, it would not be necessary to label the SEs as "small", "medium", "large"; it would be sufficient to compare them with each other, using comparative adjectives.

From this perspective, I question the authors' argument that not reporting effect size has effects on publication time. Although it may have such an effect, a statement like this may signal that the authors are not aware of the true consequences of not reporting their measure of interest. (In fact, in my personal opinion the under reporting of the effect sizes in the extant literature comes from authors who don't understand the practical relevance of this measure.)

Lines 144-152: On second thought, we may have gotten carried away with this statement. We have deleted the sentence. We still think this is the case (as a reviewer, I have on occasion requested a revision of a manuscript for this very reason). But it is true that the sentence takes the reader away from the main reason why the calculation (and interpretation) of ES indices should not be neglected.

In addition, in my field there are theoretical thresholds available, depending on the type of effect size employed. For instance, in PLS-SEM we use a Cohen-like effect size with a 0.02 minimum threshold, and several intermediary values to delimitate between small, medium and large. I trust that in health research there are thresholds too, therefore I would try to take a more in depth look in the first paragraph of section 1.2 and maybe make it more concrete.  

Taking into account this suggestion and a previous one, we have deleted the added text in lines 88-93, and we have left the beginning of section 1.2 as it was in the first version.

Personally, I am interested in your comments on the effect size labels in PLS-SEM. The same is true for SEM as for more traditional regression models. The effect size of each relationship represented in the structural model can be calculated and compared with each other (and, if available, with those of other models). The use of thresholds with arbitrary labels is not really necessary. I understand that, in working with something as specific as PLS-SEM, it can be interesting to have labels with which to quickly share results with other colleagues, but that does not replace context for interpretation. As we said in the Discussion section, a very small effect size may be relevant if the intervention on the dependent variable is trivial or at zero-cost. Cortina & Landis (2009) have one of my favorite texts on this, and they illustrate this with some examples in the social sciences. In any case, although I have never worked with PLS-SEM (only with covariance-based SEM), I would like to learn more about this topic and about arbitrary conventions for categorizing effect sizes. In the future I would like to address, precisely, the use and interpretation of ES in SEM, as a statistical tool widely used in many applied fields.

Cortina, J. M., & Landis, R. S. (2009). When small effect sizes tell a big story, and when large effect sizes don’t. In C.E., Lance, & R.J. Vandenberg (Eds.) Statistical and methodological myths and urban legends: Doctrine, verity and fable in the organizational and social sciences, (pp. 219–246.).
